# The Tripartite Dimensions of "Ren 人" (Human Beings) in Pre-Qin Confucianism in Terms of "Li 礼" (Ritual)

**Li Feng**

Department of Philosophy and Religious Studies, Peking University, Beijing 100871, China;
1901110798@pku.edu.cn

**Abstract:** This study delves into the Pre-Qin Confucian understanding of "ren 人" (human beings), focusing on the tripartite dimensions of "shen 身" (body), "qing 情" (sentiment), and "xin 心" (mind) as viewed through the lens of "li 礼" (ritual). By analyzing the works of Confucius, Mencius, Xunzi, and other significant early texts, we unravel how these early Confucian philosophers reconceptualized human beings within the framework of "li 礼" (ritual). In doing so, they presented a novel perspective on the human experience that emphasized the interconnectedness of these three dimensions, transforming the way people thought about themselves and their place in the world. This research illuminates the unique contributions of Pre-Qin Confucianism to the understanding of human beings and provides valuable insights into the philosophical breakthroughs of this period in Chinese thought. Furthermore, this understanding of human beings persisted throughout the subsequent imperial history of China.

**Keywords:** Pre-Qin Confucianism; human beings; li 礼 (ritual)





## 1. Introduction

Throughout the history of human civilization, the exploration of existential questions such as "what is a human?" and "who am I?" has never ceased. The philosophical responses to these inquiries demonstrate certain similarities across time periods and regions. In an era known as the "philosophical breakthrough" and the "axial age," the philosophical perspectives and interpretations of humanity found in regions such as ancient Greece, Israel, and India diverge from those in early China, particularly in Confucianism. So, what is the Confucian response to these questions? This article primarily addresses this topic. We believe that the most prominent and influential Chinese conception of "ren 人" (human beings) lies in the concept of "li 礼" (ritual) as advocated by early Confucianism.

Before delving deeper into our discussion, it is necessary to elaborate on why we should pay attention to this particular issue. Upon reviewing literature such as Munro's *The Concept of Man in Early China* and Xu Fuguan's 徐复观 (1903–1982) *A History of Chinese Human Nature Theory*, it becomes evident that contemporary scholars tend to focus more on aspects such as human values, morality, spiritual realms, and abstract concepts within Confucian thought. Indeed, this is a vital aspect of the concept of "human being" that emerges from the breakthrough era of philosophy. Yet, an exclusive emphasis on these dimensions can lead to an overly simplified and flattened interpretation of "ren 人" (human being) from the Pre-Qin Confucian perspective, even though it might sound captivating to perceive the essence of "ren 人" as "moral," or to attribute profound abstract ideas to "ren 人" (human being). In truth, the "human being" in the eyes of early Confucianism is much more vibrant and multifaceted. The question then arises, how do we depict this vivid "Man" in words, or which key Confucian concept aligns most closely with their understanding of "human being"?

This brings us to the second query we need to address: Why do we place an emphasis on "li 礼" (ritual) here, considering it as the pre-Qin Confucian understanding of "ren 人"

(human being)? Looking back at a wealth of crucial texts, many scholars adopt a macroscopic view in understanding "li 礼", perceiving Confucian "li 礼" (ritual) as a synonym for social order and norms[1]. As applied to the individual, it becomes an "object" for the "ren 人" (human being)—a kind of external standard, system, or even means—resembling Xin Guanglai's 信广来 instrumentalist interpretation of "li 礼" (ritual). However, to be frank, this understanding can easily invoke negative associations with "constraint, control, asceticism, hierarchy, and authority" (Z. Wang 2007, pp. 5–6), creating an impression of li 礼 as oppressive or suppressive to human nature[2]. On the other hand, viewing "li 礼" solely as an external norm or tool to mold "human being" into a moral subject risks reducing it to a limited interpretation. As Schwartz points out, "Li is not merely a set of formal rituals; it also embodies the enduring elements of the universal way".

Fortunately, many scholars have noted this "injustice" in the interpretation of "li 礼". They strive to "extract" "li 礼" from these understandings confined to specific rituals or external norms, attempting to restore its close relationship with human life and its growth process. For instance, Du Weiming 杜维明 stated, "li 礼 is the process of humanization" (Du 2002, pp. 25–26), expressing the outward display of humans in a specific environment. Ge Zhaoguang 葛兆光 proposed that we should not simplify our understanding of "li 礼", but instead focus on its source of rationality, as "li 礼" is the natural manifestation of human emotions and heavenly law (Ge 2013, pp. 52–53). Sterckx directly observed in Mencius' works that "li 礼" forms an integral part of a person's character. Roger T. Ames focuses on the interpretation of "li as body" found in ancient texts, interpreting "li 礼" as a "embodied living" (Ames 2011, p. 109). He asserted that in Confucian thought, a person's way of life is a specific form of behavioral manifestation, which is precisely defined as "li 礼" (Ames 2011, p. 113). This reminds us of the remarkable work conducted by Fingarette in interpreting early Confucian thought. He unambiguously asserts that the Analects' text (Lunyu 论语) supports and enriches our understanding of human as being, at its core, a ceremonial being.

We follow in the footsteps of Fingarette and many other scholars both at home and abroad, affirming that for early Confucianism, humans are considered ceremonial beings. This groundbreaking understanding and recognition of "human" is a result of the Confucian tradition since Confucius himself, which built upon the foundations of the Xia, Shang, and Zhou dynasties' civilizations. In contrast to the widespread belief that human existence and its justification (as a ceremonial being) primarily relied on external religious forces such as Heaven or gods during the three dynasties and earlier, the advent of the philosophical breakthrough era saw Confucius propose the "the root of ritual" (lizhiben 礼之本), signifying the shift in understanding "humans" from a focus on heavenly principles to an emphasis on the internal aspects of human life (B. Wang 2001, p. 68). For early Confucians, the manner and meaning of human existence were no longer tied to external divine destinies or related to religion; rather, they were deeply rooted in the life of the "human being" himself.

So, what exactly is this life and the levels within it? It is evident that many scholars have noticed the tendency of pre-Qin Confucians to explain "Man" through "Li", continuously emphasizing important dimensions such as life, process, and context. However, more detailed aspects have not been fully developed and discussed. The issues that this paper seeks to clarify and supplement include the following: When early Confucians encountered a living person, under what circumstances would they consider this individual a "human," what form of life does this denote, what characteristics does their acknowledged "human" possess, and what unique aspects of Chinese philosophy does this interpretation and construction of "human" reflect? We argue that the so-called "human" in early Confucianism is a person who exists ceremonially (man as a ceremonial being), and that their entire being, from the inside out, including their physical form, sentiments, and mind, manifests li 礼 (ritual).

## 2. The Ritual Bodily Dimension of Ren: Human Physical Existence from a Confucian Perspective

By examining ancient literature regarding the concepts of "human" and li 礼 (ritual), we can easily find that many texts, such as Yili 仪礼 (the Book of Etiquette and Ceremonial) and Liji 礼记 (the Book of Rites), contain a wealth of dynamic scenes. These works primarily detail the specific physical actions and behaviors required of individuals in various life situations. Furthermore, the Analects serves as a collection of the sayings and deeds of Confucius and his disciples. In more thought-focused texts such as "Mencius" and "Xunzi," there is a substantial emphasis on the importance of adhering to external physical requirements, which comprise a significant portion of their content. This is the pre-Qin Confucians' most direct portrayal of "human": Humans are directly present in the world, real, tangible, public, and observable. This presence, which can be directly seen by the naked eye, encompasses specific physical postures, movements, and actions, all referred to as "li 礼". In other words, only when an individual's shen 身 (body) first manifests or acts in accordance with "li 礼" are they considered a "ren 人" (human being) rather than merely an animal.

Let us delve into a detailed analysis using an excerpt from The Analects of Confucius in the chapter "Yan Yuan 颜渊":

> Yan Yuan asked about humaneness. The Master said, to master the self and return to ritual is to be humane. For one day master the self and return to ritual, and the whole world will become humane. Being humane proceeds from you yourself. How could it proceed from others?
>
> Yan Yuan said, May I ask how to go about this?
>
> The Master said, if it is contrary to ritual (li 礼), don't look at it. If it is contrary to ritual, don't listen to it. If it is contrary to ritual, don't utter it. If it is contrary to ritual, don't do it.
>
> Yan Yuan said, lacking in cleverness though I am, I would like, if I may, to honor these words. (Burton 2007, p. 80)[3]

In this chapter, Confucius' teaching of "mastering oneself and returning to ritual" fundamentally entails the transformation of an individual from a non-ceremonial being to a ceremonial being, thereby establishing a nexus between the body and li 礼. Confucius' specific method for achieving this metamorphosis involves refraining from hearing, seeing, speaking, or participating in activities that contravene li 礼. These four aspects correspond to human organs: Eyes, ears, mouth, hands, and feet. Within this context, the human body is further delineated into distinct physiological organs or senses, such as ears, eyes, nose, and mouth. This distinction is also present in other Confucian texts. For instance, in the "Book of Rites-Record of Music," it states "Do not permit indolence and evil to reside in your body; let your ears, eyes, nose, and mouth, as well as your mind, all act in accordance with what is right and proper." In "Mencius-Jinxin II," it reads "The mouth desires flavors, the eye beauty, the ear sounds, the nose fragrances, and the four limbs ease and comfort. These are the natural tendencies". Moreover, the Guodian bamboo slips mention "the six faculties: ears, eyes, nose, mouth, hands, and feet." However, in Confucianism, these bodily organs and their functions must be manifested through specific actions within social life and practice, serving as expressions of "ritual" (li 礼). Without adhering to these rituals, people's limbs will be uncertain of their proper placement, their ears and eyes will be unsure about what to listen to or watch, and they will lack awareness of the appropriate etiquette for greetings and interactions. In other words, a person's physical presence must embody "ritual" (li 礼) in order to be considered genuinely "human being" (ren 人)[4].

In the "Xiang Dang 乡党" chapter of the Analects, the text describes Confucius's physical presence and specific behaviors in various ritual scenarios:

> 1. In the local community, Confucius was submissive and seemed to be inarticulate. In the ancestral temple and at court, though fluent, he did not speak lightly.

2. At court, when speaking with Counsellors of lower rank he was affable; when speaking with Counsellors of upper rank, he was frank though respectful. In the presence of his lord, his bearing, though respectful, was composed.

4. On going through the outer gates to his lord's court, he drew himself in, as though the entrance was too small to admit him.

When he stood, he did not occupy the centre of the gateway;

when he walked, he did not step on the threshold.

When he went past the station of his lord, his face took on a serious expression, his step became brisk, and his words seemed more laconic.

When he lifted the hem of his robe to ascend the hall, he drew himself in, stopped inhaling as if he had no need to breathe. (Lau 1979, p. 101)

Chinese Song Dynasty scholar Zhu Xi 朱熹 had a profound understanding of Confucius, believing that the descriptions of Confucius' "physical movements, demeanor, and expressions" all adhered to li 礼 (ritual propriety). In fact, not only did Confucius emphasize the importance of ceremonial expression in one's body, but both Mencius and Xunzi also advocated that people's "facial expressions, posture, and limb movements" should externally manifest in a manner consistent with li 礼 (ritual). This aligns with Zhu Xi's 朱熹 assertion that "a person's physical movements must exhibit the 'form' of ritual", suggesting that a "human" possesses a ceremonial body.

This perspective could easily lead to the impression that li 礼 (ritual) and ren 人 (human) are seen from an instrumental viewpoint, as if li 礼 (ritual) were a tool for disciplining individuals and the ren 人 (human) or shen 身 (body) were a subordinate entity subjected to training, governance, and construction. These two would then present a binary subject–object relationship. However, we would like to clarify that in Confucian thought, li 礼 (ritual) is not an object that exists in opposition to people. In fact, for Confucians, li 礼 (ritual) serves as an interpretation of ren 人 (human being) or "existence." Humans are considered ceremonial beings, and for a person to be deemed "human," their physical form must first exist in accordance with li 礼 (ritual).

In *Confucius: The Secular as Sacred*, Fingarette directly compares the human body to a holy vessel, drawing upon a passage from the "Analects, Gong Zhi Chang 公治长":

Tzu-Kung asked: "What would you say about me as a person?"

The Master said: "You are a utensil."

"What sort of utensil?"

"A sacrificial vase of jade."

Fingarette, based on this text, explained the complex relationship between li 礼 (ritual) and "man": "The mere individual is a bauble, malleable and breakable, a utensil transformed into the resplendent and holy as it serves in the ceremony of life." (Fingarette 1972, p. 78) The dignity of a person is affirmed in their ritual behavior, as "man at his best is justified when we see that his best is a life of holy ceremony rather than of appetite and mere animal existence." (Fingarette 1972, p. 77) In this context, I follow Fingarette's interpretation, considering that early Confucianism views humans as ceremonial beings. A person is truly man only when their body manifests the form of li 礼 (ritual). Of course, Confucian understanding of ren 人 (humans) and li 礼 (ritual) goes beyond just external form.

### 3. The Ritual Sentiment Dimension of Ren: Human Sentimental Existence from a Confucian Perspective

In early Confucianism, emphasis is placed on the direct representation and expression of human physicality. The subsequent question we must explore is why a "person" should act in such a manner? What significance and value do external bodily manifestations hold? The Confucian scholars must offer a solid foundation and persuasive reasoning for these assertions. This leads us to the second aspect of Confucian discussion concerning ren 人

(human being) and li 礼 (ritual): qing 情 (sentiments)[5]. Sentiments are directly connected to the human body and encompass various levels and components.

Firstly, the human body is divided into specific sensory organs, which inherently possess desires for external objects. As Xunzi states, Human desires are such that the eye desires to be filled with beauty, the ear with sounds, the mouth with flavors, the nose with smells, and the mind with ease and satisfaction. These five desires are inescapable in the human condition (Xunzi-Wanga 王霸). According to Xunzi, once a person has a physical body, they naturally possess desires and emotions such as love and hate, joy and anger, and sorrow and pleasures. Likewise, Mencius also discussed the specific sensory desires of taste, sight, hearing, and smell in humans. He did not actually deny these innate biological instincts and desires. For instance, he stated, "Physical appearance and complexion are things given by nature" (Mencius-Jinxin I 尽心上) and "Appreciation of beauty is a universal human desire" (Mencius-Wangzhang I 万章上) This represents the initial understanding of "sentiment" by pre-Qin Confucian scholars, focusing on physiological needs and desires.

The second dimension of human sentiment extends beyond mere desires. In pre-Qin literature, "sentiment" encompasses the "joy, anger, sorrow, fear, love, hatred, and desire" mentioned in the Book of Rites (Liji 礼记), the "love, hate, joy, anger, sorrow, and pleasure" described by Xunzi, and the "grief, joy, worry, happiness, resentment, and sorrow" mentioned in Nature Derives from Mandate (Xing Zi Ming Chu 性自命出), among others. These sentiments not only include people's instinctive natural desires but also embrace moral emotions that arise from within (such as love for family members and respect for others) (Z. Wang 2011, p. 191). Pre-Qin Confucians did not advocate for "eradicating emotions and desires." Instead, they emphasized the importance of expressing one's desires and emotions in interactions with others. To illustrate the physical expression of human "desires" and "moral emotions," we can examine the account of a family member's passing as recorded in the Book of Rites (Liji 礼记).

"The distress of the heart, the painful sense of the disease, and the sorrowful sinking of the spirits, all accumulating in the breast, made it necessary to bare the breast and leap wildly, so as to move the bodily frame and compose the mind, and to allow the troubled current of the breath to subside. A woman ought not to bare her breast, and therefore she beats on her chest, and leaps with her legs apart, going on in this way as if she would break down a wall. Such is the expression of the extreme of sorrow and painful feeling." (Liji 礼记-Wensang 问丧)

As recorded in the Book of Rites (Liji 礼记), when people confront the death of a loved one, they need to engage in specific physical actions due to their overwhelming sorrow. In other words, human emotions require a release through tangible physical behaviors and expressions. Confucianism, in this context, interprets and transforms individuals' outward physical actions into representations and outlets for their inner emotional states.

Certainly, Confucianism does not advocate for a world where everyone's emotions and desires are unrestrainedly displayed. They firmly believe that unbridled indulgence in emotions and desires can only bring about tremendous disasters, such as the strong oppressing the weak, the majority suppressing the minority, the clever deceiving the honest, the brave taking advantage of the timid, the sick not receiving care, and vulnerable groups such as the elderly and orphans not receiving proper support. Therefore, Confucianism emphasizes that the expression of human sentiments should be appropriate and balanced. As Liji 礼记-Zhongyong 中庸 states, "When joy, anger, sorrow, and pleasure have not yet arisen, it is called the Mean. When they arise to their appropriate levels, it is called Harmony." Here, expressing emotions "to their appropriate levels" implies adhering to "rites" (li 礼). For instance, in funeral rites, Confucianism posits that if pounding one's chest and leaping are methods to convey grief, then the frequency of these actions should be restrained and controlled to express an appropriate level of sorrow. In this context, such measured emotional expression can be referred to as "li 礼".

According to Liji 礼记-Wensang 问丧, li 礼 do not descend from the heavens, nor do they arbitrarily spring from the earth. Instead, they represent the temperate expression of people's natural sentiments, encompassing both desires and emotions. When these sentiments are conveyed through li 礼 (ritual), they assume a moral quality. The Pre-Qin Confucian interpretation and reasoning of "human" presents a particular logic: In the eyes of Confucianism, a "person" should not only display external physical expressions but also communicate genuine internal emotions. The moderate expression of sentiments (and their manifestations) can be considered as li 礼 (ritual). In essence, Confucianism views a person as a ceremonial being with ceremonial sentiments.

## 4. The Ritual Mental Dimension of Ren: Human Mental Existence from a Confucian Perspective

However, several questions arise: With innumerable situations in the world, how can people ensure that their actions are always appropriate? What makes one's definition of appropriateness valid? Who determines what is appropriate? Where does the rationale for early Confucians' belief in the proper way of living lie? Why must people exist in such a manner? In response to these questions, Confucians provide philosophical justifications. They believe that humans have the capacity to exercise restraint when expressing emotions and to perform suitable, appropriate actions in various life situations, ensuring harmony among all things. This ability to regulate oneself (jie 节) and exhibit appropriate behavior (yi 义) is attributed to the "mind" (xin 心). As such, the early Chinese Confucian understanding of "human" does not stop at sentiment. Their ideal "human" must be able to adapt and exercise self-restraint (jie节) in their physical and emotional expressions, possessing what can be called a ceremonial xin 心 (mind).

From the Confucian perspective, humans possess a mind that exerts a dominant and controlling influence over their bodies. As stated in Guodian Bamboo Slips Five Elements Text (Wuxing 五行):

> "The ears, eyes, nose, mouth, hands, and feet are all servants of the mind. When the mind says 'yes,' none dare to refuse; when it says 'agree,' none dare to disagree; when it says 'advance,' none dare not to advance; when it says 'retreat,' none dare not to retreat; when it says 'deep,' none dare not to go deep; when it says 'shallow,' none dare not to be shallow."

Xunzi's "Uncovering the Hidden" (Jiebi 解蔽) chapter also claims that "xin 心 (mind) is the sovereign of the body and the master of spiritual clarity." "In these contexts, the "mind" can be initially understood as a unique human capacity for "rationality and discernment."

In Pre-Qin Confucianism, the concept of xin 心 (mind) encompasses not only pure rational thinking but also a clear moral orientation, often referred to as "moral virtues." These moral virtues, however, are categorized differently by various Confucian scholars. For instance, Mencius identifies four components of the "mind": ren 仁 (benevolence), yi 义 (righteousness), li 礼 (ritual), and zhi 智 (wisdom). In contrast, Guodian Bamboo Slips Five Elements Text (Wuxing 五行) divides the xin 心 (mind) into five components: ren 仁 (benevolence), yi 义 (righteousness), li 礼 (ritual), and zhi 智 (wisdom), and sheng 圣 (sageliness). Despite the classification differences, li 礼 is generally regarded as a central aspect of the human "mind" (xin 心) in Pre-Qin Confucian thought, representing a unique human "moral rational capacity."

This moral rational capacity refers to the ability to exercise self-restraint and adaptability in various situations. For example, Mencius posits that "the moral virtue of yielding marks the inception of propriety" (cirang zhixin lizhiduanye 辞让之心, 礼之端也), and "a respectful mind constitutes the essence of ritual" (gongjing zhixin liye 恭敬之心，礼也). In this context, "yielding and respect" (cirang, gongjing 辞让 恭敬) symbolize a rational capacity to modulate one's emotions and behavior when engaging with others and navigating the world. Simultaneously, this capacity is also moral in nature, as demonstrated by Liji 礼记-Sangfusizhi 丧服四制 which unequivocally states that "self-restraint is the essence of

ritual" (jiezhe liye 节者礼也). From a Confucian standpoint, individuals should express their emotions with discretion, striking a balance between excess and insufficiency while ensuring appropriateness and suitability. The ability of individuals to exhibit "restraint" and "regulation" (jie 节) when manifesting their inner emotions through physical actions, such as dance and jumping, is attributed to the presence of a ceremonial mind.

Understanding the human mind in terms of morality (or li 礼 ritual) raises the question of whether morality (li 礼) is innate or acquired. In the Pre-Qin Confucian school represented by Mencius, it is believed that humans possess a "ceremonial mind" (xin 心), characterized by the innate rational capacity for self-restraint and adaptation. However, for Xunzi, the "mind" (xin 心) primarily represents the cognitive and practical ability to understand and apply li 礼 (ritual). According to Xunzi, humans can recognize becoming a ceremonial being as the optimal choice precisely because they possess this rational capacity (i.e., the "mind" xin 心).

These contrasting perspectives within Confucianism seek to explain why humans exist in such a unique way. Regardless, the focus of this paper is not on whether the inclination for li 礼 in the mind (xin 心) is innate or acquired. For Confucians, the human mind must embody "li 礼", whether innate or acquired. On one hand, possessing a mind of "li 礼" enables individuals to become moral agents or ceremonial beings. On the other hand, a ceremonial mind refers to a state in which morality is manifested within one's psyche, reflecting their inner spiritual experiences and mental states.

It must be emphasized that the human mind (xin 心) discussed here is not the same as the "mind" within the dualistic framework of mind–body thought that has persisted in the West since Descartes. In fact, Fingarette has made an effort to abandon this interpretive framework while arguing that Confucians see humans as ceremonial beings. He continually highlights situational and dynamic behaviors, devoting substantial attention to demonstrating that Confucius' concept of li 礼 (ritual) pertains to behavior. Fingarette even introduces the concept of "The Locus of the Personal" to argue that, for Confucius, "it is action and public circumstances that are fundamental, not esoteric doctrine or subjective states."

However, this effort might be somewhat radical. Schwartz has rebutted Fingarette's assertion, stating:

> "The question here is not whether Confucius conceived of the mind-body problem in any dualistic Western way, but whether he attributes emotions, virtues, intentions, and attitudes to living individuals or somehow sees these mental phenomena as embedded only in concrete acts of li and whether he believes that the 'heart,' with all its capacities, has an autonomous, dynamic life of its own apart from specific responses to specific situations." (Schwartz 1985, p. 75)

Indeed, by persistently emphasizing the "outer," "public," "objective," and "observable" (Fingarette 1972, p. 53) characteristics of li 礼 (ritual) and humans, Fingarette inadvertently overlooks the equally present "mind" (xin 心) behind the visible world of ritual.

This article posits that the early Chinese Confucian concept of the "mind" (xin 心) is not solely about "cogito" or a kind of introverted mental cultivation, but rather a "direct response of the heart" (Tang 2016, p. 64). when one interacts with external matters and objects, which must be directly manifested in the body. This is a phenomenon of "psychosomatic merge "(shenxin ronghe 身心融合) (Ames 2006, p. 491). According to Mencius, "What is inherent in the noble person is ren 仁 (benevolence), yi 义 (righteousness), li 礼 (ritual), and zhi 智 (wisdom), rooted in the heart. Their life and development are seen in the face, appear on the back, and show themselves in the movements of the four limbs. The four limbs, without speaking, show it by their movements." In more contemporary terms, it could be said that for Pre-Qin Confucians, the mind must be "visualized" or made manifest in a tangible manner.

For early Confucians, people possess a ceremonial mind, which serves as the foundation for being a ceremonial being: First, this ceremonial mind can be directly and simultaneously manifested in one's bodily posture and behavior; second, we should also pay

attention to the guiding role of the rational dimension of the mind on sentiments and the body, as it is under the influence of the ceremonial mind that one can become a "ceremonial being," or a moral existence. If the understanding of "human being" before Confucius was still at the level of cosmological order, then, in the face of an ever-changing world, pre-Qin Confucians turned their focus to something more enduring than the cosmos: The "autonomous personality" (Voegelin 2001, p. 100) was discovered. Within the human mind (xin 心), Confucians found the basis and meaning of human existence.

**5. Conclusions**

From this discussion, it is clear that early Confucians reinterpreted and redefined ren 人 (human beings) through the lens of li 礼 (ritual). In their view, a genuine ren 人 (human being) must manifest the state of li 礼 (ritual) in their body, sentiment, and mind. As Confucians elucidated the nature of human in this manner, they also furnished reasons for why humans exist as they do. Whether it is Mencius' argument that this "represents the inherent nature of human life" (B. Wang 2023, p. 16) or Xunzi's claim that people transform into ceremonial beings based on rational choices, early Confucians did not rely on external forces, such as heaven or gods. Instead, they chose to employ their own rationality to dispel fears or illusions about the world (Ge 2013, p. 66), living and existing with courage and trust in their own human power. In summary, by concentrating on the self, early Confucians underscored the significance of human autonomy and agency, ultimately defining a true human as a ceremonial being with a harmonious fusion of body, sentiment, and mind. This philosophical breakthrough in the Chinese context presents a unique perspective on the concept of human beings. This is precisely what this paper aims to argue and present: The Tripartite Dimensions of ren 人 (human beings) in Pre-Qin Confucianism in terms of li 礼 (ritual).

By this stage, the Confucian construct of "human being," as defined prior to the Qin Dynasty, had effectively "triumphed" over its various rivals including but not limited to Daoism, Mohism, and Legalism. As we ventured into the Imperial Era, this understanding of "human being" as a ceremonial being exhibited remarkable resilience and vitality. For nearly two millennia, from the Qin and Han Dynasties through to the Qing Dynasty, this concept permeated the social fabric, evolving into a widespread self-perception and frame of reference for the Chinese populace. It morphed into a universal notion, persistently relevant despite challenges and critiques. Even though this concept was repeatedly tested and scrutinized, it consistently re-emerged victorious, its influence never once interrupted.

**Funding:** This research received no external funding.

**Institutional Review Board Statement:** Not applicable.

**Informed Consent Statement:** Not applicable.

**Data Availability Statement:** Not applicable.

**Conflicts of Interest:** The author declares no conflict of interest.

**Notes**

[1] For instance, Yu Yingshi 余英时 accentuates the significance of "li 礼", contending that the ritual traditions of the three dynasties, or "li 礼", furnished a direct historical and cultural backdrop for China's axial breakthrough. (Yu 2014, p. 16). Zheng Kai 郑开 also conducted remarkable research in this field, highlighting the political dimension of "li 礼" (see Zheng 2009). It should be underscored that this paper's contemplation and interpretation of "li 礼" are founded on previous research. Although this paper chiefly confines "li 礼" to the dimension of the individual, it nonetheless recognizes "li 礼" as an expression of order.

[2] Similarly, it should be clarified that we do not deny the dimension of "li 礼" as a tool for moral education. This perspective of "li 礼" is very evident in the philosophy of Xunzi. For instance, consider Roel Sterckx's interpretation of Xunzi's concept of "li 礼", wherein he suggests that Xunzi turned "li 礼" into a stringent rulebook that allows for the functioning of society. However, he cautiously referred to "li 礼" in Xunzi's philosophy as "second nature." (Sterckx 2022, pp. 204–5).

3  This section is translated from Burton (see Burton 2007). For all the first-hand materials in this study, the translations draw on modern Chinese and existing English versions, thoughtfully tailored or reinterpreted based on the demands of the text, context, and content.

4  However, for Daoist scholars, they did not perceive "human being" as a ceremonial being. In Zhuangzi's writings, many characters were presented with physical disabilities; despite their physical deficiencies, these individuals were depicted as having a nobler inner spiritual state. The Daoists even posited that only when humans cease using their senses can they discover the light within (see Munro 1969).

5  The Chinese character "qing 情" possesses multiple connotations within pre-Qin philosophical texts. It can denote innate natural desires inherent to human beings, signify emotional reactions at a psychological and physiological level, describe subjective feelings towards certain matters or individuals, or convey affection towards others. In this discussion, we will initially translate "qing 情" into English using the term "sentiment," bearing the broadest sense. However, throughout the ensuing discourse, we will adapt our choice of vocabulary to the most appropriate term that aligns with the context under consideration.

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
