# Peer review of "The Tripartite Dimensions of “Ren 人” (Human Beings) in Pre-Qin Confucianism in Terms of “Li 礼” (Ritual)"

_religions, doi:10.3390/rel14070891_

Round 1

Reviewer 1 Report

The research topic uses a new perspective to view Confucianism. 

The discussions in the article are valid and appropriate. However, some of the contents could be elaborated. 

1. The author should elaborate more on the literature review in the introduction section. The author should outline the current studies in the field of Pre-Qin Confucianism, the opinions or perspectives from previous scholars, and how this study could engage with current scholarships. 

2. The author should also explain more about his/her research contributions in the introduction and conclusion sections. What is something new in this study, and how does it fill the research gap? How does the research result reflect the understanding towards Pre-Qin Confucianism? Why so important to know the human being concept in Confucianism thought?

3. The arguments in the main text are good. The evidence shown came from various primary sources. But it could be better if the current discussion should be elaborated more in detail and made the analysis deeper. For example, the comparison of different Chinese schools(yinyangjia陰陽家、fajia法家、daojia道家、mojia墨家, etc) in the period towards the view of human beings(人) in their thought. Or elaborate on the change of Confucianism view after the Pre-Qin period. This could help to highlight the uniqueness of Pre-Qin Confucianism.   

4. Some of the references in the main text citations do not appear in the list of References. The author should list the version he/she referred to the primary sources such as liji禮記、mengzi孟子、xunzi荀子、zhongyong中庸, etc. In addition, the author should not overlook the work of Ge Zhaoguang葛兆光, a distinguished professor in this related field.

Author Response

Thank you so much for the opportunity to revise my manuscript entitled “The Tripartite Dimensions of "Ren人" (Human Beings) in Pre-Qin Confucianism in terms of "Li礼" (Ritual)” (manuscript number: 2457793). I sincerely appreciate the constructive comments and suggestions provided by you, which have been invaluable in improving the quality of my work.

I have studied the comments very carefully. A detailed reply to each of your comments can be found in the attached document. Please see the attachment.

Reviewer 2 Report

Please see the attached file for some corrections. Book titles are italicized and not put in "quotation marks".  Bibliography format is inconsistent and incorrect. Consult a manual of style for how to do this.

Check the spelling of "affable" (which you have rendered as "afiable").

You rely quite a bit on Fingarette, and while his work was important in its day, that "day" is now quite some time ago.  Some more recent English-language scholarship would be helpful.

See above.

Author Response

I appreciate the time and effort you've put into providing these detailed comments on my manuscript, “The Tripartite Dimensions of 'Ren人' (Human Beings) in Pre-Qin Confucianism in terms of 'Li礼' (Ritual)”, (manuscript number: 2457793). Your constructive feedback is instrumental in improving my work.

I've carefully reviewed your comments and have made the appropriate changes accordingly. Specifically, I have corrected the formatting of book titles, fixed spelling errors, and ensured that the references follow a consistent bibliographic format in line with the specified style guide.

In response to your note on my reliance on Fingarette's work, I value your input and have made necessary adjustments in the manuscript. To diversify the perspectives in my paper, I've broadened the range of references in the introductory section. Here, I've delved into the current state of both domestic and international research on Pre-Qin Confucianism's notions of 'Ren' and 'Li'. This enables me to bring to light the issues I aim to explore in my paper. To this end, the references I've drawn upon include, but are not limited to, scholars like Fingarette, Schwarz, Du Weiming, Ge Zhaoguang, Ames, and Roel Sterckx.

I trust these revisions address your concerns, and I look forward to your further comments.

Round 2

Reviewer 1 Report

I am satisfied with the author's amendment.